# Stress Responses in Horses Housed in Different Stable Designs during Summer in a Tropical Savanna Climate

**DOI:** 10.3390/ani14152263

**Published:** 2024-08-04

**Authors:** Chanoknun Poochipakorn, Thita Wonghanchao, Kanokpan Sanigavatee, Metha Chanda

**Affiliations:** 1Science and Innovation for Animal Health Program, Faculty of Veterinary Medicine, Kasetsart University Bang Khen Campus, Bangkok 10900, Thailand; chanoknun.p@ku.th; 2Thailand Equestrian Federation, Sports Authority of Thailand, Bangkok 10330, Thailand; thita.wo@ku.th; 3Department of Large Animal and Wildlife Clinical Science, Faculty of Veterinary Medicine, Kasetsart University Kamphaeng Saen Campus, Nakorn Pathom 73140, Thailand; 4Center for Veterinary Research and Innovation, Faculty of Veterinary Medicine, Kasetsart University Bang Khen Campus, Bangkok 10900, Thailand

**Keywords:** air velocity, ammonia, heart rate variability, horse, relative humidity, stable housing, temperature, tropical region, single-box, welfare

## Abstract

**Simple Summary:**

The use of single-box housing for domestic horses is a matter of concern due to its potential negative effects on their social interaction and movement. Horses in tropical regions may also face challenges due to heat and humidity fluctuations affecting their thermal comfort and heat dissipation. This study sought to explore the stress responses in horses housed in different stable designs in a tropical savanna region, as little research has considered this topic. The results reveal correlations across stable designs between internal humidity, air temperature, ammonia levels, and heart rate variability (HRV) modulation in horses. We also observed distinct stress responses in horses in different stable designs. These findings suggest that daily changes in the internal environment, as well as stable design, play crucial roles in affecting horses’ stress responses and overall well-being in tropical savanna regions. These results have important welfare implications for the management of horse housing in such environments.

**Abstract:**

Single-confinement housing can pose welfare risks to domestic horses. This study investigated horses’ stress responses when confined to single stalls in different stable designs in a tropical savanna region to address a gap in the literature. In total, 23 horses were assigned to a stable with a central corridor and solid external walls (A) (N = 8), a stable with one side corridor and solid external walls (B) (N = 6), or a stable with a central corridor and no solid external walls (C) (N = 9). Air velocity, relative humidity, air temperature, and noxious gases were measured inside the stables, and the heart rate and HRV of the horses were also determined. The relative humidity was lower in stable C than in stable A (*p* < 0.05), while the air temperature was higher in stable C than in stable B (*p* < 0.05) during the day. The airflow and ammonia levels were higher in stable C than in stables B and A (*p* < 0.01–0.0001). Overall, horses’ HRV in stable A was lower than in those in stables B and C (*p* < 0.05–0.01). Horses in stable A tended to experience more stress than those in other stables.

## 1. Introduction

Horses are social animals, and when housed in more natural conditions, such as outdoor group housing, they can exhibit species-specific behaviours [1,2], which are believed to improve their well-being [1,2,3]. However, this type of housing can also pose risks to the horses’ welfare by increasing the likelihood of injury and illness [1,4] as well as potentially reducing the human–animal bond and impairing performance when released to paddocks [5,6]. Despite the evidence of the literature highlighting the negative effects of single confinement for horses, such as limited movement [7], a lack of social interaction, and the disruption of natural feeding behaviours [8,9], many horse caretakers still opt for single-box housing due to its convenience and the desire to avoid injuries in paddocks [6,10]. Single-stall housing remains the predominant housing method globally, especially for sports and school riding horses [11], with prevalence rates ranging from 32–90% in different nations [2,12,13,14]. This housing method is associated with abnormal repetitive behaviours [15,16]. Therefore, providing appropriate care is crucial to minimise the potential stressors that could compromise the welfare of horses housed in single stalls.

To support the well-being of single-confined horses, certain countries have established recommended box sizes. For example, in the United Kingdom, the British Horse Society recommends a box size of 3.6 × 3.6 m [17]. In Sweden, the Swedish Board of Agriculture suggests a minimum area of 8.0 m^2^ with the shortest side measuring 2.35 m, for small horses, and 9.0 m^2^ with the shortest side measuring 2.5 m, for large horses [18]. In addition, according to Fédération Equestre Internationale (FEI) rules and regulations, horses participating in international equestrian events must be accommodated in boxes that are at least 3 × 3 m [19]. Despite similar confinement conditions, welfare concerns can vary with geographic and physical factors [20]. For example, in areas with reduced grazing space, horses may appear thinner and supplementary food may be necessary to maintain a healthy body condition score during warmer, drier seasons [21,22]. Horses can thrive within a wide thermal comfort zone of approximately 5 to 25 °C, depending on their ability to regulate their body temperature [23]. However, they lose their ability to dissipate heat effectively at a relative humidity (RH) above 50% [23]. Therefore, relative humidity and air temperature significantly affect stress responses in horses.

Horses are raised for specific purposes in various regions worldwide, each with its own unique climate and physical environment. These differences can lead to distinct stress responses among horses. The Köppen climate classification identifies a tropical climate as one of five major groups; it is characterised by persistently high temperatures and humidity throughout the year [24,25]. In the coldest months, the average temperature hovers around 18 °C [24]. Tropical climates are divided into three subgroups: tropical rainforest, tropical monsoon, and tropical savannah, which are distinguished by the level of precipitation in the driest month [24]. Such hot, humid conditions have a notable impact on animal physiology [26]. Temperature and humidity both powerfully affect heat dissipation in animals [27]. As humidity increases, heat loss decreases, while an increase in temperature leads to an increase in heat loss [27,28,29]. Furthermore, high temperatures can cause animals to experience heat stress, which is worsened by high relative humidity due to reduced heat loss via evaporative cooling [30]. Poor housing conditions can exacerbate these conditions, causing thermal stress and related diseases in animals in the humid tropics [31]. These stress conditions could affect animal production and compromise animal welfare in tropical environments [25,32].

Numerous works in the literature discuss how horses respond to stress in basic housing in temperate regions. However, there is a lack of information on these stress responses in tropical regions, specifically in tropical savanna climates where weekly relative humidity and air temperature fluctuations are a common concern. Furthermore, there is little understanding of how different barn types affect horses’ stress response in tropical savanna climates. This study aims to investigate and compare stress responses via the modification of heart rate (HR) and heart rate variability (HRV), which are reliable parameters for indicating stress in horses [33,34,35,36], during 24 h housing in three different types of barns in a specific area located in a tropical savanna climate. This will test the hypothesis that significant variations in a horse’s stress response occur depending on the type of barn used in tropical savanna climates.

## 2. Materials and Methods

### 2.1. Animals

For this study, 23 healthy horses (12 geldings, 1 stallion and 10 mares) were selected. Of those, 15 horses were provided by the Horse Lover’s Club, Pathum Thani, Thailand, while 9 were recruited from the House of Horses Riding Club, Bangkok, Thailand. They were housed in standard horse stables containing separate boxes with straw bedding. Commercial pellets were provided at a rate of 1.5–2 kg total daily, divided equally into three portions. Pangola hay and water were supplied ad libitum. The horses were regularly trained for four days per week for local jumping competitions or school riding and were given a regular day off on Monday. The horses in this study were not given medical or surgical treatment for 30 days before the study began. If the horses had shown signs of clinical illness during the 24 h of data collection, they would have been excluded from the study. However, no horses were removed, so the HR and HRV can be analysed from all 23 horses in this study.

### 2.2. Experimental Protocols

This study was conducted at the Horse Lover’s Club, Pathum Thani (latitude: 13.99433, longitude: 100.68079), and the House of Horses Riding Club, Bangkok (latitude: 13.81283, longitude: 100.78692), Thailand, both situated in tropical savanna climate areas [37,38,39]. Horses selected for heart rate (HR) and heart rate variability (HRV) measurements were randomly chosen from their respective stables where they were permanently housed. This was to minimise potential stress from unfamiliar housing conditions that could affect the horses’ stress responses. In addition, the experiment was performed on three consecutive Mondays, the club’s day off, in April 2024, when the temperature was above 40 °C. This scheduling was to reduce any potential influence from the clubs’ routine activities on the horses’ stress responses. A haematological examination was undertaken before each experiment day to confirm a normal range of values in the study horses. 

### 2.3. Stable Designs and Characteristics

The study involved housing horses in one of three different stable designs for 24 h straight: (1) Eight horses (four geldings and four mares, aged 20.8 ± 5.9 years and weighing 422.5 ± 57.9 kg) were housed in a 4-metre-wide centre aisle design stables with solid exterior walls (50 × 15 × 4 m) at the Horse Lover’s Club (Stable A). The exterior wall was constructed with bricks on the lower half up to approximately 170 cm in height and insulated with a layer of plaster. Wire mesh was installed in the upper half and permanently covered with a 32 squares/inch vector-protection net to prevent midges from entering. Inside, there were 20 independent 4 × 4 m boxes partitioned by solid walls to 170 cm in height with multiple vertical bars above to enable visual contact between neighbouring horses; (2) six horses (two geldings and four mares, aged 20.0 ± 2.3 years and weighing 448.5 ± 85.5 kg) were housed in a stable (50 × 7 × 4 m) that was fully covered with 32 squares/inch vector-protection net and contained only one row of 10 separate 4 × 4 m boxes at the Horse Lover’s Club (Stable B). The 3-metre-wide stable aisle was located in front of the boxes, with gates at each end that were perpendicular to the corridor. Adjacent boxes were partitioned with solid lower wall segments and upper segments of vertical metal bars to enable visual contact between neighbouring horses; (3) nine horses (six geldings, one stallion, and two mares, aged 17.4 ± 4.7 years and weighing 411.0 ± 109.9 kg) were housed in a 4-metre-wide centre aisle barn with no exterior walls (40 × 12 × 4 m) at the House of Horses Riding Club (Stable C). The stable contained two rows of horse boxes and was temporarily covered with a tiny mesh net at night. The stable’s corridor was between the rows of boxes with the front and back gates at each end, parallel to the corridor. The tiny mesh net was rolled up to allow air to flow into the horse boxes during the day. In all stable designs, the front and back gates were opened during the day and completely closed from 18:00 to 05:30. Damp straw bedding and manure were removed twice daily at 05:30 and 17:30 on the experiment dates in all three stable designs.

### 2.4. Data Acquisition

#### 2.4.1. Humidity and Air Temperature

Humidity and air temperature during the experiment were determined with three sets of temperature and humidity data logger devices (TM-305U; Tenmars Electronics, Taipei, Taiwan). Two devices were positioned in the stable corridor (Figure 1f, Figure 2e and Figure 3f), and one outside the stable. The average humidity and air temperature inside the stables were established as a percentage of relative value (%) and degrees Celsius (°C). Data were exported at 1 min intervals and reported as average values of each consecutive 60 min for 24 h.

#### 2.4.2. Air Velocity and Noxious Gases

Ammonia (NH_3_) levels were evaluated inside the stables with two portable ammonia gas monitoring devices (SC-04 (NH_3_); Riken Keiki, Tokyo, Japan). Two pieces of the portable gas monitoring device (GX-3R; Riken Keiki, Tokyo, Japan) were utilised to measure various gases within the stables, including oxygen (O_2_), methane (CH_4_), carbon monoxide (CO), and hydrogen sulphide (H_2_S). The gas monitoring devices were placed inside each of the stables at heights of 60–70 cm from the floor. Two airflow anemometers (GM8902, BENETECH, Shenzhen, China) were put inside and outside the stables facing in the same direction to assess the internal and external wind speeds during the 24 h of the experiments. The NH_3_, CO, and H_2_S levels were reported as parts per million (ppm), while O_2_ and CH_4_ were described as a per cent and per cent of the lower explosive limit (LEL), respectively. Finally, the airflow was determined as m/s. Data were exported at 1 min intervals and reported as average values of each consecutive 60 min for 24 h.

#### 2.4.3. Heart Rate and Heart Rate Variability

Heart rate (HR) and heart rate variability (HRV) were employed to measure the horses’ stress responses while they were housed in given stable designs. Heart rate monitoring (HRM) devices (Polar Electro Oy, Kempele, Finland) were used to record beat-to-beat (RR) interval data and, subsequently, the HR and HRV analyses. The HRM devices are valid and produce reliable HRV results in horses [40,41,42]. A set of HRM devices consists of a Polar equine belt for riding, a heart rate sensor (H10), and a Polar sports watch (Vantage V3). On the experimental days, the horse was equipped with the HRM device 30 min before recording began to accustom it to the instrument. The Polar belt attached to the sensor was soaked in water before ultrasound gel was applied to enhance the transmission signal. The belt was then fastened to the horse’s chest and the sensor pocket was placed on the left side of the chest. Finally, the sensor was synchronised and wirelessly connected to the sports watch for continuous RR recording from 07.00 h at the beginning of the experimental day to 07.00 h on the subsequent day.

The RR interval data from the Polar sports watch were uploaded to the Polar FlowSync program (https://flow.polar.com/, accessed on 12 May 2024) and exported as CSV files, which were used to compute HRV variables with Kubios premium software (Kubios HRV Scientific; https://www.kubios.com/hrv-premium/, accessed on 12 May 2024). The results were reported as MATLAB MAT files. Automatic artefact detection was used to correct or exclude artefacts and ectopic beats in the RR interval time series. Automatic noise detection was set at a medium level to mark noise segments, which can distort several consecutive beat detections and, thus, affect the accuracy of the HRV analysis. All segments marked as noise were excluded from the HRV analysis. Trend components were adjusted using the smoothness priors at 500 ms. The cutoff frequency for trend removal was set at 0.035 Hz as outlined by the user guideline (https://www.kubios.com/downloads/Kubios_HRV_Users_Guide.pdf, accessed on 12 May 2024).

The HRV variables were computed via three methods, as shown in Table 1. The frequency band thresholds were based on the recommendations of Kuwahara et al. (1996) [43]. Due to routine stable cleaning at 05.30–06.30 h, which may have distorted the HRV variables, the HRV data from 05.00–07.00 h were excluded from the analysis. In addition, the horses were routinely fed pellets at 05.00 h, 11.00 h, and 17.00 h and may have shown arousal behaviours then. Accordingly, the HRV variables were determined at 1 h intervals from 07.00 h on the experimental day to 05.00 h on the following day, and the RR interval data were excluded for about 20 min from the analyses, at 10.50–11.10 h and 16.50–17.10 h, to avoid a distorted HRV estimation. 

### 2.5. Statistical Analysis

GraphPad Prism version 10.2.3 (GraphPad Software Inc., San Diego, CA, USA) was used for the statistical analysis. Due to missing data on HRV analysis, the independent effects of stable type and time and the interaction effect of stable type-by-time on changes in relative humidity and air temperature, airflow, and HR and HRV variables were assessed via the mixed-effects model (the restricted maximum likelihood: REML) with Greenhouse–Geisser correction. Tukey’s post-hoc test was utilised for within-group and between-group comparisons at specific times. The Shapiro–Wilk test was used to evaluate the normal distribution of the data when necessary. Due to the non-normal distribution of the data, the Kruskal–Wallis test, followed by Dunn’s multiple comparisons test, was implemented to estimate variations in the horses’ age and weight and the ammonia levels in the stables; in addition, Spearman’s rank correlation (*r_s_*) was employed to estimate the correlations among relevant parameters. The correlation coefficients were defined as weak (±0.10 ≤ *r_s_* < ±0.40), moderate (±0.40 ≤ *r_s_* < ±0.70), strong (±0.70 ≤ *r_s_* < ±0.90), or very strong (*r_s_* ≥ ±0.90) [44]. The results are expressed as mean ± SD, with statistical significance at *p* < 0.05.

## 3. Results

### 3.1. Relative Humidity and Air Temperature

The graphs in Figure 4 illustrate the fluctuations in relative humidity and air temperature outside and inside the stables over 24 h. Humidity decreased while air temperature increased during the day, both outside (Figure 4a,b) and inside the stables (Figure 4c,d). Conversely, humidity increased while air temperature decreased at night both outside and inside the stables (Figure 4a–d).

During the day, lower external humidity and higher external air temperature were recorded than at night in the three stable designs (humidity, stables A and B, *p* < 0.05; stable C, *p* < 0.001; air temperature, stables A and B, *p* < 0.01; stable C, *p* < 0.0001) (Figure 5a,b). However, differences in internal humidity and air temperature between day and night were only observed in stable C (humidity: *p* < 0.001; air temperature: *p* < 0.0001) (Figure 5c,d). Furthermore, stable C’s internal humidity was lower than stable A’s during the day (*p* < 0.05) (Figure 5c), but its internal air temperature was higher than that of stable B during the day (*p* < 0.05) (Figure 5d).

### 3.2. Air Velocity and Noxious Gases 

Figure 6a,b illustrate the external and internal air velocity over 24 h. The external air velocity did not vary between day and night among the three stables. However, during both day and night, stable C exhibited a higher external air velocity than stables A and B (*p* < 0.0001) (Figure 6c). Additionally, the internal air velocity in stable C was greater during the day than at night (*p* < 0.05) and surpassed that of stables A and B during both the day and night (*p* < 0.0001) (Figure 6d).

No internal NH_3_ levels were detected throughout the day, but they increased significantly at night, reaching a peak at 03.00–06.00 h in the three stable designs (Figure 7a). At night, NH_3_ levels were higher in stable C than in stables A and B (*p* < 0.01 for both) (Figure 7b). H_2_S, NH_3_, and CO were not detected inside the stables, while O_2_ levels remained constant at 20.9% over 24 h.

### 3.3. Heart Rate and Heart Rate Variability

In this study, horses of similar ages (stable A: 20.8 ± 5.9 years; stable B: 20.0 ± 2.3 years and stable C: 17.4 ± 4.7 years, *p* > 0.05 for all comparison pairs) and weights (stable A: 422.5 ± 57.9 kg; stable B: 448.5 ± 85.5 kg and stable C: 411.0 ± 109.9 kg, *p* > 0.05 for all comparison pairs) were assigned to three stable designs, where they were housed for 24 h. The HR and HRV variables were computed as measures of autonomic regulation with the following methods.

#### 3.3.1. Time Domain Method 

Group–by–time interactions and independent group and time effects were detected with the RR triangular index modification (*p* = 0.0085, *p* = 0.0230 and *p* < 0.0001). The effects of group–by–time and time were observed as modulations in mean HR (*p* = 0.0370 and *p* < 0.0001), mean RR intervals (*p* = 0.0427 and *p* < 0.0001), and SDANN (*p* = 0.0416 and *p* < 0.0087). The effects of group-by-time and group were detected via the SDNN variable (*p* = 0.0297 and *p* = 0.0311). A sole group–by–time interaction affected changes in RMSSD (*p* = 0.0496), while group alone exerted an effect on changes in TINN (*p* = 0.0468) and pNN50 (*p* = 0.0476) (Figure 8 and Figure 9). 

The mean HR was lower at 9.00–11.00 h and 00.00–02.00 h compared to 17.00–18.00 h in horses in stable A (*p* < 0.05 for all). It was also lower in horses in stable B at 00.00–01.00 h compared to 21.00–22.00 h (*p* < 0.01). In horses in stable C, the mean HR decreased at 03.00–05.00 h compared to 13.00–14.00 h (*p* < 0.01) and at 02.00–04.00 h compared to 17.00–18.00 h (*p* < 0.05) (Figure 8a). The mean RR intervals were significantly higher in stable A than in stable B at 08.00–09.00 h (*p* < 0.05). In horses in stable A, the mean RR intervals were lower at 16.00–19.00 h than at 09.00–10.00 h (*p* < 0.05 for all); they were also lower at 15.00–22.00 h than at 01.00–02.00 h (*p* < 0.05 for all). In stable C, the RR intervals were higher at 01.00–05.00 h than at 17.00–18.00 h (*p* < 0.05 for all) and lower at 7.00–8.00 h and 13.00–14.00 h compared to 03.00–04.00 h (*p* < 0.05 and *p* < 0.01). There was no variation in mean RR intervals in horses in stable B throughout the 24 h (Figure 8b). 

TINN was lower in stable A than in stable B at 18.00–19.00 h (*p* < 0.05) and in stable C at 20.00–21.00 h (*p* < 0.01), 01.00–02.00 h (*p* < 0.05), and 03.00–05.00 h (*p* < 0.05 for both) (Figure 8c). The RR triangular index was higher in stable B than in stables A and C at 07.00–08.00 h (*p* < 0.05 for both comparison pairs), 08.00–09.00 h (A vs. B, *p* < 0.01 and B vs. C, *p* < 0.05), 11.00–12.00 h (*p* < 0.01 for both comparison pairs), 12.00–13.00 h (A vs. B, *p* < 0.01 and B vs. C, *p* < 0.05), 15.00–16.00 h (A vs. B, *p* < 0.01 and B vs. C, *p* < 0.05), and 18.00–19.00 h (A vs. B, *p* < 0.001 and B vs. C, *p* < 0.01). Furthermore, the RR triangular index was higher in stable B than in stable A at 17.00–18.00 h (*p* < 0.05), and it decreased at 19.00–05.00 h compared to 18.00–19.00 h in horses in stable B (Figure 8d).

SDNN was lower in stable A than in stable B at 9.00–10.00 h, 11.00–12.00 h, 17.00–18.00 h, and 22.00–23.00 h (*p* < 0.05 for all times) and stable C at 9.00–10.00 h (*p* < 0.05), 20.00–00.00 h (*p* < 0.05 for all times), 01.00–02.00 h (*p* < 0.01), and 03.00–04.00 h (*p* < 0.05). The SDNN decreased at 17.00–18.00 h compared to 07.00–08.00 h in stable A (*p* < 0.05) but increased at 21.00–22.00 h compared to 12.00–13.00 h in stable B (*p* < 0.05) (Figure 9a). The SDANN in stable B was lower than that in stable A at 07.00–08.00 h (*p* < 0.05) but higher than that in stable C at 20.00–21.00 h (*p* < 0.01). The SDANN in stables A and B was lower than that in stable C at 23.00–00.00 h (*p* < 0.05 for both comparison pairs) (Figure 9b). The RMSSD in stable A was lower than that in stable B at 18.00–19.00 h (*p* < 0.05) and stable C at 9.00–10.00 h, 17.00–18.00 h, 20.00–21.00 h, 1.00–2.00 h, and 3.00–4.00 h (*p* < 0.05 for all times). The RMSSD decreased at 13.00–14.00 h compared to 08.00–09.00 h in stable A (*p* < 0.05) (Figure 9c). The pNN50 in stable A was lower than that in stable B at 9.00–10.00 h, 11.00–12.00 h, and 17.00–18.00 h and stable C at 20.00–21.00 h, 22.00–23.00 h, and 01.00–02.00 h (*p* < 0.05 for all times). The pNN50 was higher at 21.00–22.00 h compared to 12.00–13.00 h in stable C (Figure 9d).

#### 3.3.2. Frequency Domain Method

The group–by–time interaction affected changes in the LF band (*p* = 0.0403), HF band (*p* = 0.0403), LF/HF ratio (*p* = 0.0216), total power (*p* = 0.0285), and RESP (*p* = 0.0142). The VLF band was affected independently by the group (*p* = 0.0064) and time (*p* < 0.0001) (Figure 10).

The VLF band in stable A was higher than that in stable B at 7.00–10.00 h, 12.00–14.00 h, and 18.00–19.00 h and higher than that in stable C at 20.00–21.00 h and 03.00–04.00 h (*p* < 0.05 for all times). The VLF band increased at 03.00–04.00 h compared to 11.00–12.00 h in stable A (Figure 10a). The LF band was higher in stable A than in stables B (*p* < 0.05) and C (*p* < 0.01). The LF band rose at 04.00–05.00 h compared to 17.00–18.00 h in stable A (*p* < 0.05). It also increased at 04.00–05.00 h compared to 09.00–10.00 h in stable B (*p* < 0.05) (Figure 10b). In contrast to the LF band, the HF band was lower in stable A than in stables B (*p* < 0.05) and C (*p* < 0.01). The HF band dropped at 04.00–05.00 h compared to that at 17.00–18.00 h in stable A (*p* < 0.05). It also declined at 04.00–05.00 h compared to that at 09.00–10.00 h in stable B (*p* < 0.05) (Figure 10c). The total power was lower in stable A than stable C at 09.00–10.00 h (*p* < 0.01), 21.00–22.00 h (*p* < 0.05), 23.00–00.00 h (*p* < 0.05), and 01.00–02.00 h (*p* < 0.01) (Figure 10d). The LF/HF ratio was higher in stable A than in stables B (*p* < 0.01) and C (*p* < 0.05). The LF/HF ratio rose at 04.00–05.00 h compared to that at 09.00–10.00 h in stable B (*p* < 0.01) (Figure 10e). The RESP was higher in stable A than in stable C at 20.00–21.00 h (*p* < 0.05). It also increased at 20.00–21.00 h compared to that at 15.00–16.00 h in stable C (*p* < 0.05) (Figure 10f).

#### 3.3.3. Nonlinear Method and Autonomic Nervous System Index

Group–by–time interactions affected SD1 modulation (*p* = 0.0496), while group affected SD2 modulation independently (*p* = 0.0299). The effects of group–by–time (*p* = 0.0478) and time (*p* < 0.0001) influenced changes in the PNS index, but the SNS index changed primarily due to time alone (*p* < 0.0001) (Figure 11).

SD1 was lower in stable A than in stable B at 18.00–19.00 h (*p* < 0.05) and stable C at 9.00–10.00 h, 17.00–18.00 h, 20.00–22.00 h, 01.00–02.00 h, and 03.00–04.00 h (*p* < 0.05 for all times). The SD1 was lower at 13.00–14.00 h than at 08.00–09.00 h (*p* < 0.05) (Figure 11a). The SD2 was also lower in stable A than in stable C at 9.00–10.00 h, 20.00–22.00 h, 23.00–00.00 h and 01.00–02.00 h (*p* < 0.05 for all times) (Figure 11b). The PNS index in stable C was higher than that in stable A at 17.00–18.00 h (*p* < 0.05) and stable B at 02.00–03.00 h and 04.00–05.00 h (*p* < 0.05 for both times). In stable A, the PNS index decreased at 17.00–20.00 h compared to that at 09.00–10.00 h (*p* < 0.05–0.01). It then increased at 0.00–02.00 h compared to 17.00–18.00 h (*p* < 0.05 for both comparison pairs). The PNS index in stable C was lower at 7.00–8.00 h and 12.00–14.00 h compared to 02.00–03.00 h (*p* < 0.05 for all comparison pairs) (Figure 11c). The SNS index peaked at 17.00–18.00 h compared to 07.00–11.00 h (*p* < 0.05 for all comparison pairs) and 19.00–05.00 h (*p* < 0.05–0.0001) (Figure 11d).

### 3.4. Correlation between HRV Variables and Internal Environment

The correlation matrix of HRV variables and internal environmental measurements, including relative humidity, air temperature, NH_3_ levels, and airflow in the three stable designs, are shown in the Appendix A. Internally in the three stables, the relative humidity was negatively correlated with air temperature in stables A (r_s_ = −0.96), B (r_s_ = −1.00), and C (r_s_ = −0.95) but positively correlated with their NH_3_ levels (r_s_ = 0.68, 0.65, and 0.84, respectively). Likewise, the air temperature was negatively correlated with ammonia in stables A (r_s_ = −0.71), B (r_s_ = −0.65), and C (r_s_ = −0.89). In stables A and C, airflow showed a positive correlation with air temperature (stable A, rs = 0.41; stable C, r_s_ = 0.56) but a negative correlation with NH_3_ levels (stable A, r_s_ = −0.60; stable C, r_s_ = −0.41). Airflow was also correlated with the relative humidity in stable C (r_s_ = −0.47). No correlation was found between airflow and other internal environment measures in stable B.

In stable A, the relative humidity, air temperature, and NH_3_ levels show a moderate to strong correlation with HR (r_s_ = −0.72, r_s_ = 0.63, r_s_ = −0.56) and various HRV variables, specifically, RR intervals (r_s_ = 0.72, r_s_ = −0.66, r_s_ = 0.59), SDANN (r_s_ = 0.73, r_s_ = −0.60, r_s_ = 0.47), TINN (r_s_ = 0.66, r_s_ = −0.61, r_s_ = 0.53), VLF (r_s_ = 0.57, r_s_ = −0.55, r_s_ = 0.44), LF (r_s_ = 0.64, r_s_ = −0.71, r_s_ = 0.47), HF (r_s_ = −0.64, r_s_ = 0.71, r_s_ = −0.47), PNS index (r_s_ = 0.76, r_s_ = −0.68, r_s_ = 0.60), and SNS index (r_s_ = −0.82, r_s_ = 0.74, r_s_ = −0.60) (Appendix A).

According to the measurements taken in stable B, the relative humidity and air temperature show a moderate correlation with HR (r_s_ = −0.50, r_s_ = 0.51) and several HRV variables, specifically, RR intervals (r_s_ = 0.51, r_s_ = −0.52), SDANN (r_s_ = 0.47, r_s_ = −0.47), pNN50 (r_s_ = −0.43, r_s_ = 0.43), TINN (r_s_ = 0.44, r_s_ = −0.42), RR triangular index (r_s_ = −0.45, r_s_ = 0.46), VLF (r_s_ = 0.49, r_s_ = −0.49), LF (r_s_ = 0.63, r_s_ = −0.61), HF (r_s_ = −0.63, r_s_ = 0.61), LF/HF ratio (r_s_ = 0.60, r_s_ = −0.57), and SNS index (r_s_ = −0.48, r_s_ = 0.49). Meanwhile, NH_3_ levels were moderately correlated with HR (r_s_ = −0.61), RR (r_s_ = 0.62), SDANN (r_s_ = 0.55), pNN50 (r_s_ = −0.45), RR triangular index (r_s_ = −0.57), VLF (r_s_ = 0.49), PNS index (r_s_ = 0.47), and SNS index (r_s_ = −0.55) (Appendix A).

The relative humidity, air temperature, and NH_3_ levels show a moderate to strong correlation with HR (r_s_ = −0.66, r_s_ = 0.70, r_s_ = −0.70) and various HRV variables in stable C, specifically, RR intervals (r_s_ = 0.64, r_s_ = −0.69, r_s_ = 0.69), RMSSD (r_s_ = 0.51, r_s_ = −0.54, r_s_ = 0.63), pNN50 (r_s_ = 0.45, r_s_ = −0.49, r_s_ = 0.53), TINN (r_s_ = 0.56, r_s_ = −0.51, r_s_ = 0.62), SD1 (r_s_ = 0.51, r_s_ = −0.54, r_s_ = 0.63), PNS index (r_s_ = 0.69, r_s_ = −0.75, r_s_ = 0.73), and SNS index (r_s_ = −0.68, r_s_ = 0.73, r_s_ = −0.73) (Appendix A).

## 4. Discussion

This research analysed the stress responses of horses housed in various stable designs during the summer in a tropical savanna climate. The main findings are as follows: (1) internal humidity and temperature differed across three stable designs, despite similar variations in the external environment; (2) changes in these environmental factors were connected to the regulation of ammonia, which increased with higher nighttime humidity; (3) unrestricted airflow of high-temperature air increased temperature and decreased humidity inside stables without solid external walls during the day; (4) horses exhibited the highest SNS index at 17:00–18:00 h, while horses in stables with solid exterior walls showed a reduced RR interval, coinciding with an increased HR at the same time; (5) changes in various heart rate variability (HRV) parameters were associated with shifts in the internal environment and decreased when the horses were housed in stables with solid exterior walls. These findings suggest that the design of stables can impact the stress responses of horses during summer in a tropical savanna climate.

The tropical savanna climate is one of the climate categories designated in the Köppen climate classification [45]. It is characterised by distinct wet and dry seasons and is primarily found in Africa, Asia, Central America, and South America [37]. The tropical zone receives more direct solar radiation, resulting in higher humidity and ambient temperatures compared to other regions of the world [46,47]. In addition, annual fluctuations in both the maximum and minimum ambient temperatures can affect farm animals in tropical areas, creating stress [37,48]. The most common mistake in modern horse stable construction is poor ventilation [49]. A closed housing system is recommended for areas with high rainfall and temperate climates [50]. Conversely, in tropical climates, housing facilities that allow for natural ventilation are best suited for promoting optimal animal health [47,50]. Therefore, the housing systems that optimise health and welfare conditions should be selected according to the specific local climate. In this investigation of summer housing, we observed daily fluctuations in external humidity and temperature at different times of day and night at the three stable locations. However, notably, significant day–night differences within stables were only evident in stable C. During the day, decreased humidity coincided with increased temperature inside stable C, mirroring the outside conditions. In stable C, we also observed higher internal air velocity during the day, which was associated with the measured internal humidity and temperature. The design of stable C, which lacks solid external walls, may allow freer air circulation, resulting in higher air temperature due to circulation from the external environment. This, in turn, raised the stable’s internal temperature and lowered its internal humidity. 

Aside from air velocity, we also discovered correlations between internal humidity, temperature, and ammonia levels across all three stable designs, indicating the influence of these environmental parameters on ammonia level regulation within stables. These results aligned with those reported in the previous literature on the impact of ventilation on harmful gases in dairy cattle and horse stables [51,52]. Furthermore, internal ammonia levels rose alongside increasing internal humidity overnight in all three stable designs. This observation aligns with a previous report demonstrating increased ammonia levels that correspond to the high humidity and low air velocity at night in free-stall dairy barns [53]. In addition, ammonia levels are correlated with air temperature when straw bedding is utilised in horse housing [54]. Reportedly, the sample collection location significantly affects NH_3_ compound detection within horse stables [55]. To avoid this confounding factor, the gas measuring devices for this study were placed at a constant height during the experiments across comparable stable designs. In this study, the peak ammonia levels at 03.00–06.00 h in each stable design were consistent with a previous report describing the highest ammonia levels at 04.00 h [55], suggesting that overnight is a critical period of ammonia accumulation within horse stables. The recommended maximum ammonia levels of ≤10 ppm [51] in a dairy house and ≤20 ppm in horse stables are suggested to ensure optimal health and welfare [52]. As the internal ammonia levels in this study (0–4 ppm) did not reach this limit, we assume that horse welfare was not affected by them in these three stable designs. The greatest ammonia levels were observed in stable C, although no variation in internal humidity and temperature at night was found across the three stable designs. The floor slope, which was thought to aid in the drainage of urine and contaminated fluid, was lower in stable C (3.13 ± 0.16°) compared to stables A (4.98 ± 0.24°) and B (4.98 ± 0.18°), which may have affected the measured ammonia levels. Moreover, each box contained less straw bedding in stable C than in other stables. This allowed such fluids to flow slowly and be absorbed into the concrete floor, facilitating ammonia emission within the boxes. This was thought to result in earlier ammonia detection and a higher measured ammonia level in stable C than in other stables. Together, these findings emphasise the critical role of stable design in shaping the internal environment in a tropical savanna climate.

In practical terms, one way to assess horses’ stress responses is by observing various biological parameters, such as behavioural changes [56,57,58], hormonal release [59,60], biochemical variables [61], and autonomic nervous system (ANS) function [34,36,62,63,64,65]. ANS regulation can be estimated via HRV, which is described as the fluctuation in the time interval between adjacent heartbeats under the influences of the sympathetic and vagal nerves within the cardiac cycle [66,67,68]. This natural phenomenon reflects the body’s adaptive capacity to handle external challenges and maintain bodily homeostasis [69,70]. Changes in multiple HRV variables reflect autonomic regulation, for example, reduced SDNN, RMSSD, HF band, and SD1 mirror short-term variations in heart rate, reflecting decreased vagal activity in horses [71,72]. Conversely, increased LF/HF and SD1/SD2 ratios, which are sympathovagal balance indicators [66,67], indicate an increased sympathetic role during exercise [34,36,65]. In this study, we used HRV analysis to measure stress responses as HRV can be measured noninvasively and does not interfere with horse locomotion. Interestingly, the modulation in HRV variables varied between horses, indicating distinct autonomic responses to being housed in three stable designs. Furthermore, various HRV variables (SDNN, RMSSD, pNN50, TINN, SD1, and SD2) were lower, particularly at night, in stable A than in stable C. They were also lower, to a lesser extent, in stable A than in stable B. In addition, an increased heart rate (HR), coinciding with decreased RR intervals, reflected a reduced variation in heart rate at 17:00–20:00 h, suggesting a critical period in horses in stable A. As decreased HRV mirrors a reduced role in vagal activity in stressful conditions [66,67,73], we assumed that horses experienced more stress while housed in stable A than in the other stable designs. More importantly, the correlation between changes in multiple HRV variables and internal humidity and temperature indicated the strong effect of the internal stable environment on autonomic regulation and, in turn, stress responses in horses across the three stable designs. The study results align with previous reports demonstrating the benefit of open housing facilities in preventing the adverse effects of tropical climates on animal welfare [50,51]. In addition, proper housing construction materials and ventilation are key elements of optimal housing in tropical climates [51]. Although variations in the internal environment and horses’ stress responses were observed while they were housed in different stables during summer in a tropical savanna climate, the effects of the condition in other seasons on those variables require further investigation.

The primary limitation of this study was the variation in external air velocity at different locations, resulting in differences in the internal air velocity among the three stable designs. In addition, while there were no inter-group variations in the age and weight of the horses, changes in HRV could have been influenced by individual horses’ characteristics and differences in the socialisation of the horses between the three stables. Therefore, any comparison of HRV among stable designs should be interpreted with caution.

## 5. Conclusions

The internal environment in horse stables varied depending on their designs. These variations led to different HRV changes and, therefore, different stress responses in horses housed in each of the three stable designs. Horses in stables with solid exterior walls experienced more stress than those housed in stables of other designs. These findings offer valuable insights into horses’ welfare when housed in various stable designs during the summer in a tropical savanna climate. The information from this study could contribute to developing appropriate stable management practices to minimise horses’ stress and improve their welfare in tropical regions.

## Figures and Tables

**Figure 1 animals-14-02263-f001:**
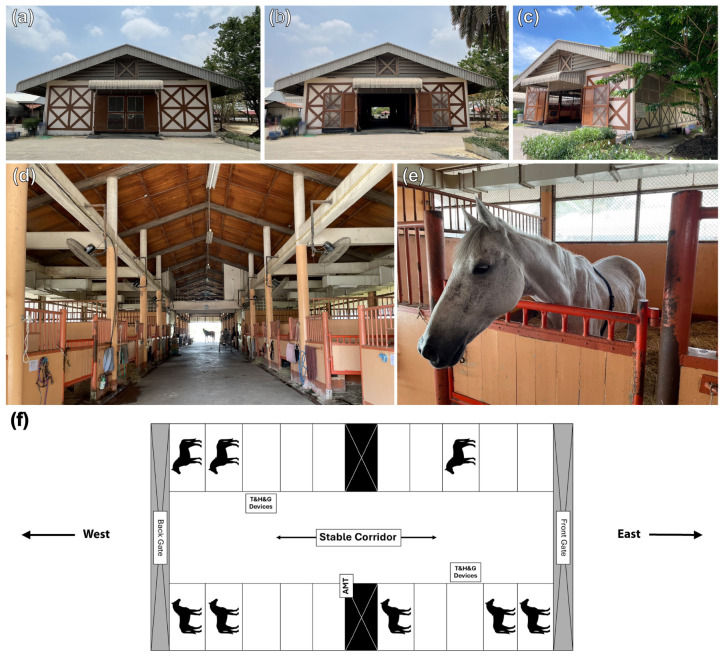
Stable A, a stable with solid external walls. The stable contains two rows of horse boxes and is permanently enclosed with a tiny mesh net (**a**–**c**). The stable corridor runs between the rows of boxes with the front and back gates at each end, parallel to the corridor (**d**). The solid portions of the barn’s external wall are designed to be taller than the horses’ heights (**e**). The RR intervals of eight horses were randomly recorded within the stable (**f**). Digital devices to measure air temperature (T), relative humidity (H), internal gases (G), and airflow (AMT) were installed inside the stable.

**Figure 2 animals-14-02263-f002:**
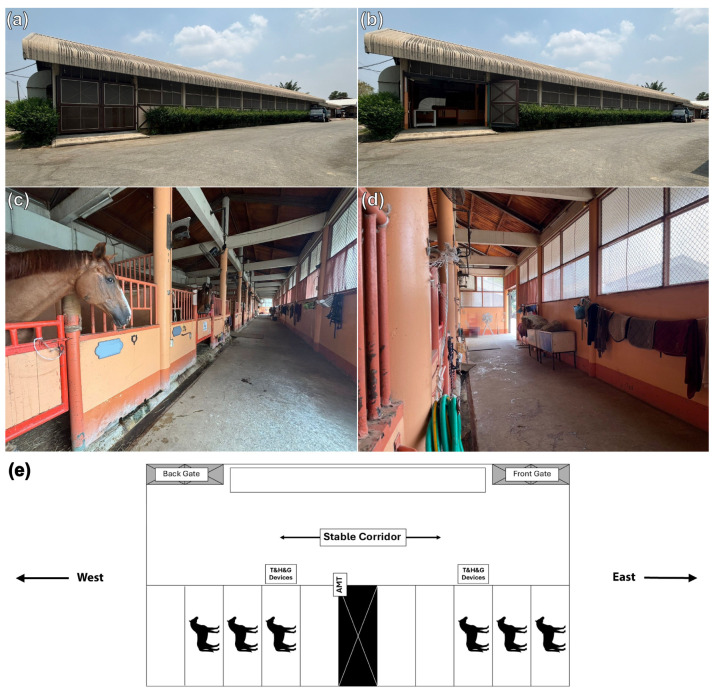
Stable B, a horse stable with solid external walls. The stable contains a row of horse boxes and is permanently enclosed with a tiny mesh net (**a**,**b**). The stable corridor runs in front of the boxes with the front and back gates at each end perpendicular to the corridor (**a**–**c**). The barn’s external wall is designed to be taller than the horses’ heights and is covered with a tiny mesh net above the solid portion (**d**). The RR intervals of six horses were randomly recorded within the stable (**e**). Digital devices to measure air temperature (T), relative humidity (H), internal gases (G), and airflow (AMT) were installed inside the stable.

**Figure 3 animals-14-02263-f003:**
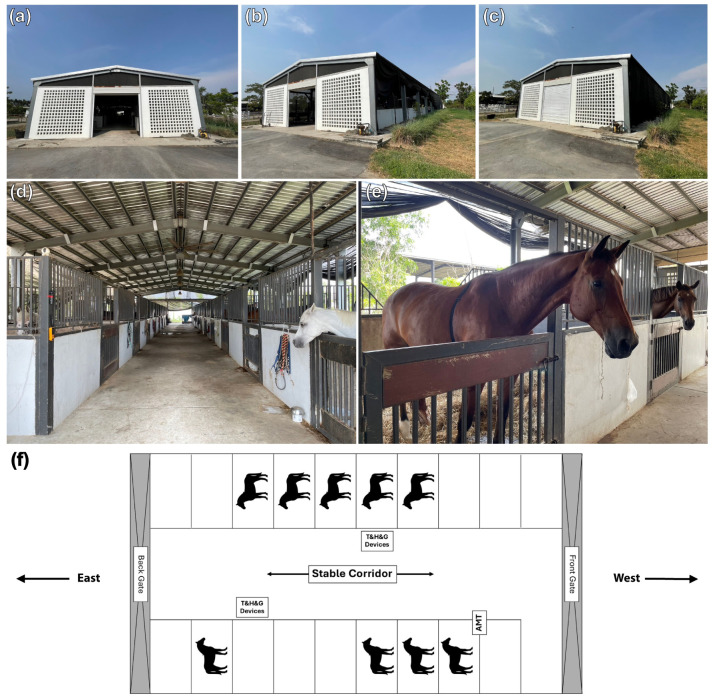
Stable C, a horse stable without solid external walls. The stable contains two rows of horse boxes and is temporarily covered with a tiny mesh net at night (**a**–**c**). The stable corridor runs between the rows of boxes with the front and back gates at each end, parallel to the corridor (**d**). The tiny mesh net is rolled up to allow natural air flow into the horse boxes during the day (**e**). The RR intervals of eight horses were randomly recorded within the stable (**f**). Digital devices to measure air temperature (T), relative humidity (H), internal gases (G), and airflow (AMT) were installed inside the stable.

**Figure 4 animals-14-02263-f004:**
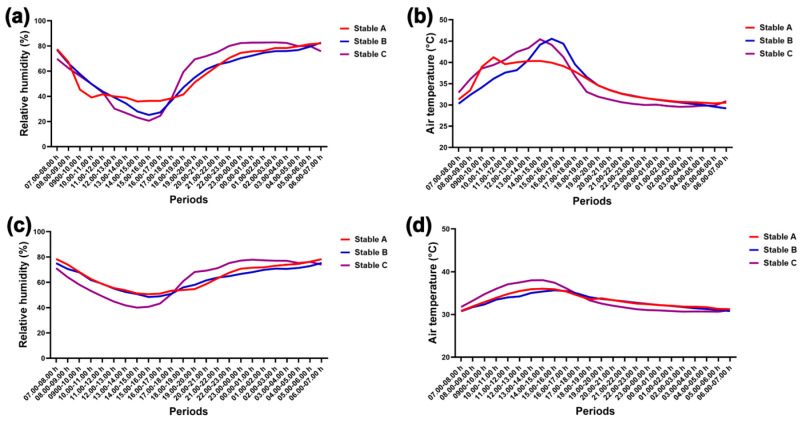
Relative humidity and air temperature outside (**a**,**b**) and inside stables (**c**,**d**) over 24 h in three stables.

**Figure 5 animals-14-02263-f005:**
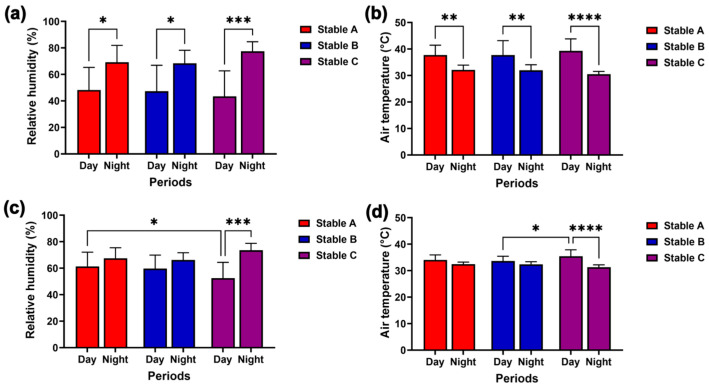
Comparison of relative humidity and air temperature among different stable designs and between day and night outside (**a**,**b**) and inside stables (**c**,**d**). *, **, ***, and **** indicate statistical significance between pairs of comparison at *p* < 0.05, 0.01, 0.001, and 0.0001, respectively.

**Figure 6 animals-14-02263-f006:**
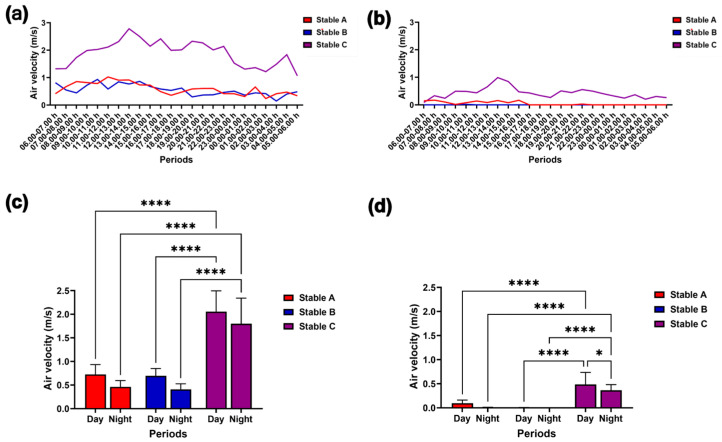
Modulation in external (**a**) and internal air velocity (**b**) during 24 h. The air velocity was compared among different stable designs and between day and night outside (**c**) and inside the stables (**d**). * and **** indicate statistical significance between pairs of comparison at *p* < 0.05 and 0.0001, respectively.

**Figure 7 animals-14-02263-f007:**
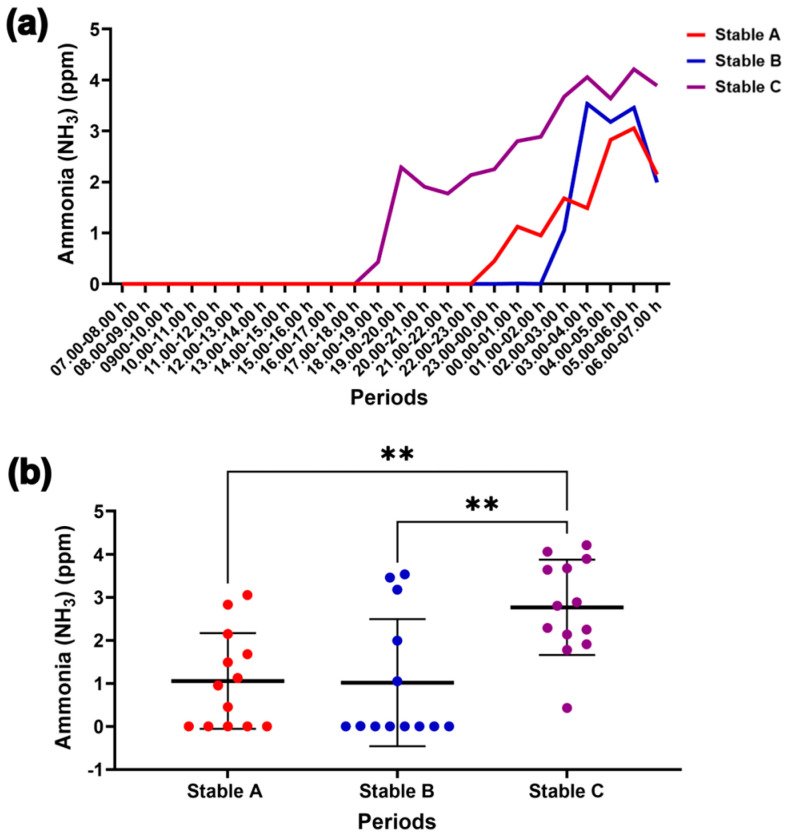
Internal ammonia level for 24 h in the three stables (**a**). Comparison of the changes in the ammonia levels among the different stable designs at night (**b**). ** indicates statistical significance between pairs of comparison at *p* < 0.01.

**Figure 8 animals-14-02263-f008:**
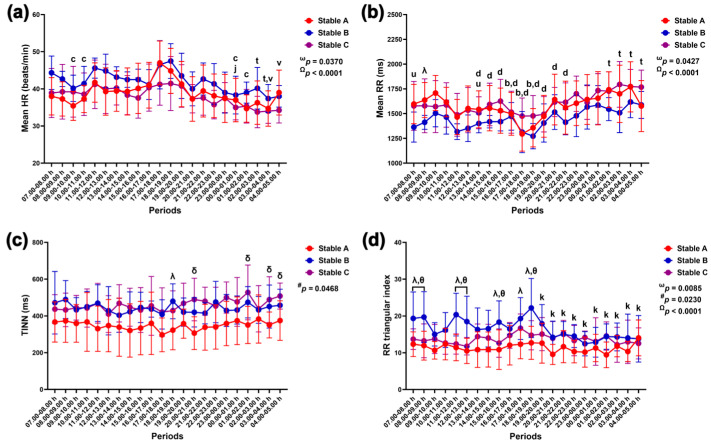
Mean heart rate (HR) (**a**), mean beat-to-beat (RR) interval (**b**), triangular interpolation of normal-to-normal intervals (TINN) (**c**), and RR triangular index (**d**) in horses during 22 h of measurement in three stables. ω, #, and Ω indicate the effects of group-by-time, group, and time, respectively. λ, δ, and θ indicate significant differences between stable A and B, A and C, and B and C at given times. b indicates a significant difference in stable A at the given times compared to the value at 9.00–10.00 h, c indicates a significant difference in stable A at the given times compared to the value at 17.00–18.00 h, d indicates a significant difference in stable A at the given times compared to the value at 01.00–02.00 h, j indicates a significant difference in stable B at the given times compared to the value at 21.00–22.00 h, k indicates a significant difference in stable B at the given times compared to the value at 18.00–19.00 h, t indicates a significant difference in stable C at the given times compared to the value at 17.00–18.00 h, u indicates a significant difference in stable C at the given times compared to the value at 03.00–04.00 h, and v indicates a significant difference in stable C at the given times compared to the value at 13.00–14.00 h.

**Figure 9 animals-14-02263-f009:**
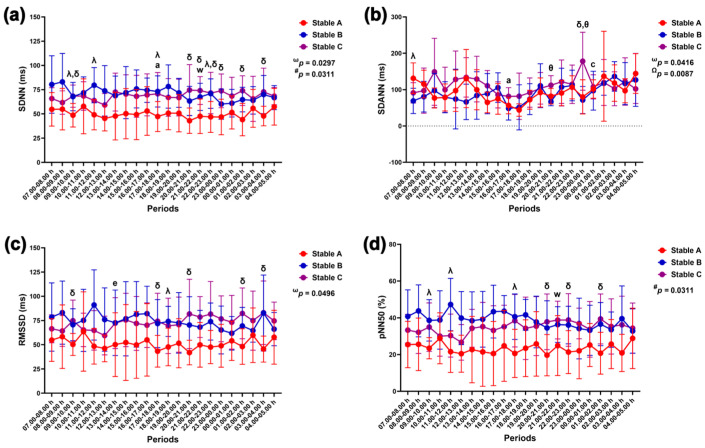
Standard deviation of normal-to-normal RR intervals (SDNN) (**a**); standard deviation of the averages of RR intervals in 5 min segments (SDANN) (**b**); root mean square of successive RR interval differences (RMSSD) (**c**); and relative number of successive RR interval pairs that differ by more than 50 ms (pNN50) (**d**) in horses during 22 h of measurement in three stables. ω, #, and Ω indicate the effects of group-by-time, group, and time, respectively. λ, δ, and θ indicate significant differences between stable A and B, A and C, and B and C at given times. a indicates a significant difference in stable A at the given times compared to the value at 7.00–8.00 h, c indicates a significant difference in stable A at the given times compared to the value at 17.00–18.00 h, e indicates a significant difference in stable A at the given times compared to the value at 8.00–9.00 h, and w indicates a significant difference in stable C at the given times compared to the value at 12.00–13.00 h.

**Figure 10 animals-14-02263-f010:**
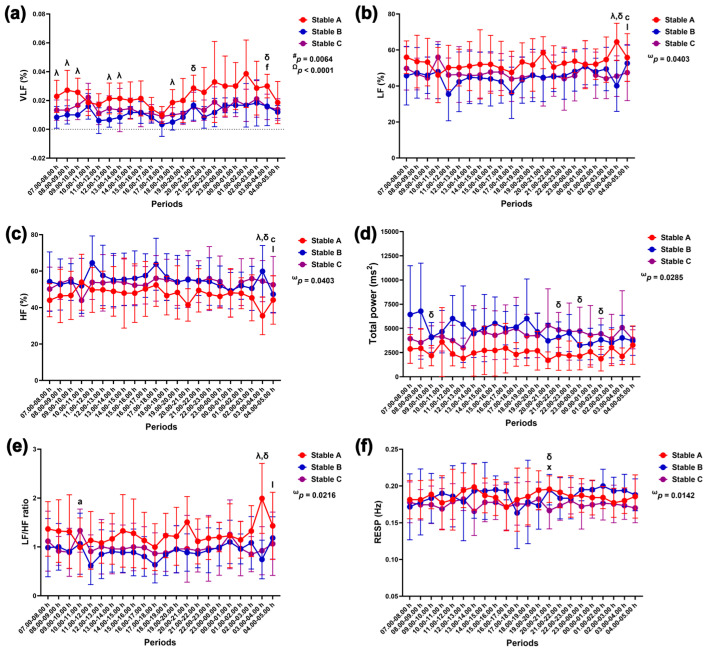
Very-low-frequency (VLF) band (**a**), low-frequency (LF) band (**b**), high-frequency (HF) band (**c**), total power (**d**), LF/HF ratio (**e**), and respiratory rate (RESP) (**f**) in horses during 22 h of measurement in three stables. ω, #, and Ω indicate the effects of group-by-time, group, and time, respectively. λ and δ indicate significant differences between stable A and B and stable A and C at given times. a indicates a significant difference in stable A at the given times compared to the value at 07.00–08.00 h, c indicates a significant difference in stable A at the given times compared to the value at 17.00–18.00 h, f indicates a significant difference in stable A at the given times compared to the value at 11.00–12.00 h, l indicates a significant difference in stable B at the given times compared to the value at 09.00–10.00 h, and x indicates a significant difference in stable C at the given times compared to the value at 15.00–16.00 h.

**Figure 11 animals-14-02263-f011:**
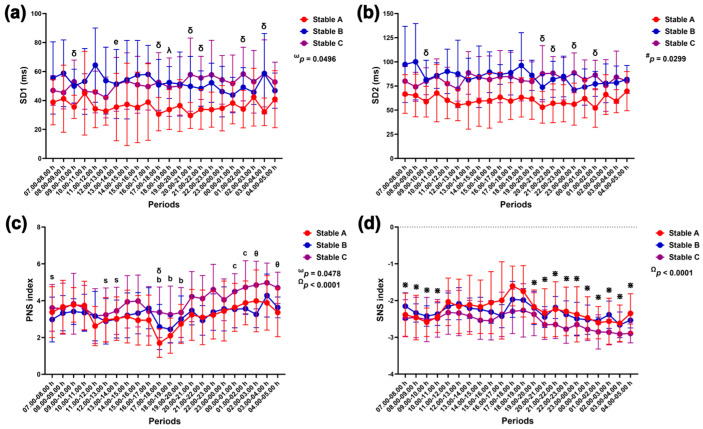
Standard deviation of the Poincaré plot perpendicular to the line of identity (SD1) (**a**) and along the line of identity (SD2) (**b**), parasympathetic nervous system (PNS) index (**c**), and sympathetic nervous system (SNS) index (**d**) in horses during 22 h of measurement in three stables. ω, #, and Ω indicate the effects of group-by-time, group, and time, respectively. λ, δ, and θ indicate significant differences between stable A and B, A and C, and B and C at given times. b indicates a significant difference in stable A at the given times compared to the value at 9.00–10.00 h, c indicates a significant difference in stable A at the given times compared to the value at 17.00–18.00 h, e indicates a significant difference in stable A at the given times compared to the value at 8.00–9.00 h, s indicates a significant difference in stable C at the given times compared to the value at 02.00–03.00 h, and * indicates a significant difference at the given times compared to the value at 17.00–18.00 h.

**Table 1 animals-14-02263-t001:** HRV variables measured in horses housed in different types of barns.

Variables	Unit	Description
Time domain method		
Mean HR	beats/min	Mean heart rate
Mean RR	ms	Mean beat-to-beat interval
SDNN	ms	Standard deviation of normal-to-normal RR intervals
SDANN	ms	Standard deviation of the averages of normal-to-normal RR intervals in 5-min segments
RMSSD	ms	Root mean square of the successive differences between RR intervals
pNN50	%	Number of successive RR interval pairs that differ by more than 50 ms (NN50) divided by the total number of RR intervals
TINN	ms	Triangular interpolation of normal-to-normal intervals computed from the baseline width of a histogram displaying RR intervals
RR triangular index	-	Integral of the density of the RR interval histogram divided by its height
Frequency domain method		
VLF	%	Relative power of the very-low-frequency band (frequency band threshold 0.00–0.01 Hz)
LF	%	Relative power of the low-frequency band (frequency band threshold 0.01–0.07 Hz)
HF	%	Relative power of the high-frequency band (frequency band threshold 0.07–0.6 Hz)
Total power	ms^2^	Total power of the spectrum band
RESP	Hz	Respiratory rate estimated from RR interval recordings
LF/HF ratio	-	The ratio of frequency band analysis for indicating sympathovagal balance
Nonlinear method		
SD1	ms	Standard deviation of the Poincaré plot perpendicular to the line of identity
SD2	ms	Standard deviation of the Poincaré plot along the line of identity
SD2/SD1 ratio	-	The ratio of nonlinear analysis for indicating sympathovagal balance
Autonomic nervous system index		
PNS index	-	Parasympathetic nervous system index
SNS index	-	Sympathetic nervous system index

## Data Availability

The data that support the findings of this study are available at https://www.doi.org/10.6084/m9.figshare.25876369 (accessed on 19 May 2024). The other information not provided in the repository platform is available from the corresponding authors, Metha Chanda and Kanokpan Sanigavatee, upon reasonable request.

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
