# Peer review of "Stress Responses in Horses Housed in Different Stable Designs during Summer in a Tropical Savanna Climate"

_animals, 2024, doi:10.3390/ani14152263_

Round 1
Reviewer 1 Report
Comments and Suggestions for Authors
The proposed manuscript is certainly interesting. Climate change is affecting the world and some countries may be ideal laboratories to study the effects of climate change on animal welfare. The manuscript describes differences in cardiac variability in horses kept in three stables with different external wall structures. Cardiac variability is an excellent tool for understanding how the autonomic nervous system responds to environmental stimuli. In theory, all environmental stimuli can induce changes in cardiac variability. The actions of people, stable routines, exercise, food, the presence of horses or other animals in the area and many other environmental stimuli can alter autonomic nervous system activity and the sympathetic-vagal balance. Individual experience can also cause one animal to have different autonomic nervous system activations than another when faced with the same environmental stimulus.
For these reasons, the manuscript needs considerable work before it can be considered for publication. Let us begin with an analysis of the study group. Although I am aware that experimental designs with horses often require methodological compromises, in this case the manuscript lacks important information (and consequently several variables were not evaluated).
The authors used a group of 23 horses from 2 stables. This group of horses was divided into 3 groups (should the authors describe the division criterion, random?). It is necessary to describe the structure of the 2 stables of origin of the horses in order to know if they are similar to one of the 3 stables where the experiment took place. It is obvious that this variable can have a decisive influence on the results. The acclimatisation period in the new stables must also be reported.
Therefore, the stud farm of origin should be included in the statistics as a dependent variable, together with: experimental stables (as a whole), gas, age and sex of the horses. In this respect, I believe that a mixed model statistic is more appropriate.
In order to improve the understanding of the experimental design, precise photographs of the outer wall and the other walls taken from the inside of the boxes should be provided. In the photographs provided, the detail of the outer wall structure is not clearly visible.
Regarding the parameters of cardiac variability, first a general consideration. The gold standard for calculating HRV is the ECG trace and it is now very easy to obtain this data, so I do not understand why we generally continue to use heart rate monitors built with human algorithms. I know it is the practice in this and many other articles, but the HRV reduces the accuracy of the data collected (see Parker et al, 2010). However, in a recent paper, the same authors (Huangsaksri et al., 2024) report the following frequencies: VLF 0-0.04 Hz, LF 0.04-0.15 Hz, HF 0.15-0.4 Hz, indicating that they are provided by default by the device. In the present work, however, the frequencies used are different: VLF 0.00-0.01 Hz, LF 0.01-0.07 Hz, HF 0.07-0.6 Hz, and again the authors state that these frequencies are provided by default by the system. The authors should justify their choice of frequencies and provide an explanation. They should also provide the raw data, either as an attachment to the paper or in a repository with a link in the manuscript. I also recommend removing the table in the supplementary materials, which has no relevance.
In addition, the discussion should take into account possible alternative explanations. For example, it can be seen that in stable C the horses have full social contact with their stable mates, which does not seem to be the case in the other stables. In my opinion, this could justify variations in HRV. In any case, it is a scientific and conceptual fallacy to believe that variations in HRV can only be attributed to variations in the structure of the external walls of the stable. Considering that so many stimuli can alter the activity of the autonomic nervous system, sentences such as the one in the conclusions "These variations led to different HRV changes and, therefore, different stress responses in horses housed in each of the three stable designs. Horses housed in solid wall stables experienced more stress than those housed in stables of other designs' are purely speculative and not scientifically relevant. Given the importance that this type of study can have, I would advise the authors to amend the MS by considering possible alternative explanations.
Author Response
Title: Stress responses in horses housed in different stable designs during summer in a tropical savanna climate
Dear Reviewer 1,
We deeply appreciate reviewer’s dedicated time and effort in reviewing our work. We’ve addressed all reviewer’s points and highlighted them in green. We also highlighted them in yellow in case they were mentioned in the text.
Reviewer 1’s comments
The proposed manuscript is certainly interesting. Climate change is affecting the world and some countries may be ideal laboratories to study the effects of climate change on animal welfare. The manuscript describes differences in cardiac variability in horses kept in three stables with different external wall structures. Cardiac variability is an excellent tool for understanding how the autonomic nervous system responds to environmental stimuli. In theory, all environmental stimuli can induce changes in cardiac variability. The actions of people, stable routines, exercise, food, the presence of horses or other animals in the area and many other environmental stimuli can alter autonomic nervous system activity and the sympathetic-vagal balance. Individual experience can also cause one animal to have different autonomic nervous system activations than another when faced with the same environmental stimulus.
Response to the reviewer
I’d like to thank the reviewer for reviewing our work and for giving us valuable comments.
For these reasons, the manuscript needs considerable work before it can be considered for publication. Let us begin with an analysis of the study group. Although I am aware that experimental designs with horses often require methodological compromises, in this case the manuscript lacks important information (and consequently several variables were not evaluated).
- The authors used a group of 23 horses from 2 stables. This group of horses was divided into 3 groups (should the authors describe the division criterion, random?). It is necessary to describe the structure of the 2 stables of origin of the horses in order to know if they are similar to one of the 3 stables where the experiment took place. It is obvious that this variable can have a decisive influence on the results. The acclimatisation period in the new stables must also be reported. Therefore, the stud farm of origin should be included in the statistics as a dependent variable, together with: experimental stables (as a whole), gas, age and sex of the horses. In this respect, I believe that a mixed model statistic is more appropriate.
Response to the reviewer
We do apologise for giving unclear information. In the present study, horses with similar age and weight ranges (already mentioned on page 3 in the Stable Designs and Characteristics subsection and page 11 in the Heart Rate and Heart Rate Variability subsection) were randomly selected from each stable design where they were permanently housed. This was done to minimise the confounding factor from individual and potential stress from unfamiliar housing conditions that could impact the horses’ stress responses. So, no acclimatisation is required in horses in each stable design. We’ve revised the context on page 3 (experimental protocol subsection) to make it more precise information.
Regarding your recommendation on appropriate statistical analysis, we appreciate your expertise and notifying us of this issue. We apologise for giving you the wrong information. In fact, the mixed-effects model (REML) with Greenhouse–Geisser correction has been used due to the missing HRV data for statistical analysis in this study, as shown in the example picture of the PNS index variable below.
We’ve revised the statistical analysis section to give more precise information on page 8 in the statistical analysis subsection.
In order to improve the understanding of the experimental design, precise photographs of the outer wall and the other walls taken from the inside of the boxes should be provided. In the photographs provided, the detail of the outer wall structure is not clearly visible.
Response to the reviewer
We’d like to thank the reviewer for this comment. Figures 1d and e of stable A and 2c and d of stable B represent the inner wall of the stall within Stable A. In the meantime, Figures 2c and d were taken inside stable B, demonstrating the lower segment of the solid external wall and the upper segment of the wire mesh and tiny mesh net. Moreover, the partition with a solid wall at the lower segment and vertical bar at the upper segment were also shown in Figures 1d and e and Figures 2c and d. We believe that the illustrating figures would convince the reader to understand the stable designs in this study. Please correct me if I am wrong.
Regarding the parameters of cardiac variability, first a general consideration. The gold standard for calculating HRV is the ECG trace and it is now very easy to obtain this data, so I do not understand why we generally continue to use heart rate monitors built with human algorithms. I know it is the practice in this and many other articles, but the HRV reduces the accuracy of the data collected (see Parker et al, 2010). However, in a recent paper, the same authors (Huangsaksri et al., 2024) report the following frequencies: VLF 0-0.04 Hz, LF 0.04-0.15 Hz, HF 0.15-0.4 Hz, indicating that they are provided by default by the device. In the present work, however, the frequencies used are different: VLF 0.00-0.01 Hz, LF 0.01-0.07 Hz, HF 0.07-0.6 Hz, and again the authors state that these frequencies are provided by default by the system. The authors should justify their choice of frequencies and provide an explanation. They should also provide the raw data, either as an attachment to the paper or in a repository with a link in the manuscript. I also recommend removing the table in the supplementary materials, which has no relevance.
Response to the reviewer
We’d like to thank the reviewer for this comment. We agree with the reviewer that ECG monitoring is the gold standard for HRV analysis. However, the heart rate monitoring (HRM) device is alternatively used in horses in the field study as it is easy to use and has been validated to provide reliable HRV results in horses (https://doi.org/10.1016/j.jevs.2021.103716, https://doi.org/10.1016/j.jveb.2014.07.006, https://doi.org/10.1016/j.jevs.2016.07.006, https://doi.org/10.1016/j.jevs.2016.07.006). The HRM device is recently used, other than the works from our group, for HRV analysis in horses (https://doi.org/10.3389/fvets.2023.1305873, https://doi.org/10.1016/j.jveb.2022.11.008, ). Although some argue that HRV variables derived from the HRM device are affected by artefacts (https://doi.org/10.1016/j.applanim.2015.02.007), this can be corrected by using the artefact correction algorithm, which detects and removes the extra or misaligned beat detections as well as ectopic beats such as premature ventricular contractions (PVC) or other arrhythmias. In the present study, we applied the Kubios HRV scientific, the premium version of the HRV analysis program that provides automatic artefact correction and noise detection algorithms. This program not only corrects the distorted beat-to-beat interval but also eliminates the physiological 2-degree atrioventricular (AV) block, generating more accurate HRV variables in horses.
Considering the reviewer’s concern about the frequency threshold, we apologise for the wrong information being given. The band thresholds of the frequency domain analysis in this study were not provided by the program as a default. Instead, they were adopted from Bowen and Marr (1998) (https://doi.org/10.1016/j.applanim.2015.02.007), who validated these thresholds in pharmacological experiments of ANS blockage in horses (https://doi.org/10.1016/0165-1838(96)00028-8) and have been used in a number of studies. We’ve revised it on page 7 in the frequency domain method of table 1. However, the default band thresholds, which are based on the human respiratory rate, have also been used in various studies in horses (https://doi.org/10.1016/j.jevs.2010.12.007, https://doi.org/10.1016/j.jveb.2019.05.003) as the horse’s respiratory rate (8-16 tpm) is quite similar to that in human (12-18 tpm). The raw HRV data of horses in this study are available at https://www.doi.org/10.6084/m9.figshare.25876369 on the Data Availability Statement on page 18. We’ve also removed Tables S1-S4 from the supplementary file as per the reviewer’s suggestion.
In addition, the discussion should take into account possible alternative explanations. For example, it can be seen that in stable C the horses have full social contact with their stable mates, which does not seem to be the case in the other stables. In my opinion, this could justify variations in HRV. In any case, it is a scientific and conceptual fallacy to believe that variations in HRV can only be attributed to variations in the structure of the external walls of the stable. Considering that so many stimuli can alter the activity of the autonomic nervous system, sentences such as the one in the conclusions: "These variations led to different HRV changes and, therefore, different stress responses in horses housed in each of the three stable designs. Horses housed in solid wall stables experienced more stress than those housed in stables of other designs' are purely speculative and not scientifically relevant. Given the importance that this type of study can have, I would advise the authors to amend the MS by considering possible alternative explanations.
Response to the reviewer
We’d like to thank the reviewer for this comment. We acknowledge the importance of social contact for horses in stable C and its potential impact on their stress levels. However, we were unable to find evidence supporting this claim in our study. Instead, we observed that horses in stable A exhibited reduced heart rate variability (HRV), indicating higher stress levels compared to those in stables C and B. Our study also revealed a connection between the internal environment (relative humidity and temperature) and the stress response of horses within different stable designs. Interestingly, horses in stable B showed similar HRV patterns to those in stable C, but with higher HRV levels, especially in the morning, suggesting they may have experienced more comfort than those in stable A. This could be attributed to the different ventilation capacities of the stables and the impact of internal pollution relative to the number of horses housed within. Based on our findings, we continue to support the idea that the modulation of the internal environment, influenced by specific stable designs, significantly affects HRV and leads to different stress responses in horses. Please correct me if I am wrong.

Reviewer 2 Report
Comments and Suggestions for Authors
Anyone who has housed horses in individual stalls has experienced the lack of air movement, but also the lack of direct sunlight in horse barns, which alter the thermal comfort of the animals. A thorough study of temperature, humidity and ammonia levels to help define best stable designs is beneficial, particularly to horses housed in more extreme temperate zones.
Although the authors provide specific details on stable designs, little information was provided regarding the horses. Did the horses remain in their original stable and stalls, and were pair matched by age/ size between the 3 barn types, or were they selected, randomized, and then moved into their stable / stalls based on randomization? If there were moved, describe their acclimatization to their new environment. How were the specific stalls that the authors used in the study selected (e.g. figures 1, 2 and 3) or did the selected horses simply stay housed in their permanent stalls??
The authors provided very specific details on the stable design, which added confusion for the reader. Certainly the details are needed, but it would be better to eliminate the word ‘standard’ to describe all 3 designs. It would be beneficial and allow the reader to visualize the barns if the authors used architectural language. Stable A was a center aisle design with solid exterior walls, B was a closed straight shed row, and C was a center aisle barn with open or no exterior walls. Then the authors could provide more details in their descriptions. Using words like small mesh should be avoided. Define the screen sizes by citing the # holes/inch or centimeter which will adequately define the screen mesh for duplication in subsequent studies.
Why didn’t the authors position any humidity devices inside the stalls?
It is recognized that HR and HRV are stress indicators, however the way the authors have presented the data is tedious and difficult for the reader to follow and interpret. Would it be better to state overall statistical effects by stable and summarize the time variations observed in the study? Alternatively would a table be a better way to present the heart rate data?? The results presented as correlations between the heart rate data and environmental factors was more informative. The table in the supplement seems unnecessary. One or two sentences in the text would be sufficient.

Comments on the Quality of English LanguageThe quality of the English was fine, however the authors need to condense the description of the stables and figure out a better way to present the heart rate and HRV data as the wording is excessive, lengthy and confusing.
Author Response

(The authors gave the same response as above.)

Reviewer 3 Report
Comments and Suggestions for Authors
The research topic is original due to difficult climat conditions of Thailand and the welfare of horses.
The study was properly design, but methodology should be improved. First, it is necessary to specify what climatic conditions prevail in the considered region, because specifying the zone and maximum air temperatures is insufficient. Please add information about the statistical climate: temperature, air humidity, wind speed and direction, rainfall, etc.
Additionally, there are concerns about keeping horses in stalls for 24 hours. It is natural to take horses out to the paddock or pasture, and constant stay in stalls may cause behavioral changes resulting in horses' anxiety. Such a change could result in a disturbed heartbeat measurement result. Please comment on how this was taken into account in the study design.
The discussion does not take into account some of the results, especially the hourly time frames. Why are the results shown in such detail if the discussion is general? It should also be supplemented with more citations. I propose to present the results more clearly, because it is difficult to verify all the markings in the text. Maybe a tabular summary would help systematize the data?
The discussion and summary do not contain any recommendations for stable users in this specific climate. Having three different building structures and measurement results, you can make recommendations, e.g. which building would be the most advantageous solution.
Author Response
In our study, we investigated the stress responses of horses in different stable designs by using various HRV variables. We acknowledge the reviewer's comment about not considering some of the results, particularly the hourly time frames, due to the limited fluctuation in just 4 out of 14 variables throughout the day. We found that horses in stable A exhibited reduced HRV compared to those in stables B and C, indicating higher stress levels in stable A. We have incorporated the feedback by including relevant information about hourly time frames, especially for horses in stable A, in the discussion section. In the results section, we utilized a mixed effect model to examine the individual effects of stable design and time, as well as the interaction effect of stable design and time on changes in HRV variables. We found distinct HRV modulation in horses housed in different stable designs, which is discussed in detail in the first paragraph of the discussion section. We further aimed to present the overall HR and HRV modulation over a 24-hour period through graph illustrations, highlighting the reduction in several HRV variables in horses in stable A compared to those in stables B and C. We believe that graph illustrations effectively convey which stable design benefits or disadvantages the horses.

Round 2
Reviewer 1 Report
Comments and Suggestions for Authors
The authors have improved the manuscript considerably. I would only suggest two small changes. The first relates to the frequency bands. Those included in this MS are the correct ones and it would be appropriate to cite Kuwahara et al (1996 Assessment of autonomic nervous function by power spectral analysis of heart rate variability in the horse. Journal of the autonomic nervous system, 60(1-2), 43-48). In terms of discussion, the authors should consider the possibility that the results obtained are also influenced by other variables, such as the difference in socialisation of the horses between the 3 stables. This strengthens the result by providing guidance to other researchers and shows that the authors have sufficient knowledge of other possible variables. A few lines in the discussion are sufficient.
Author Response
Reviewer 1
Title: Stress responses in horses housed in different stable designs during summer in a tropical savanna climate
Comments and Suggestions for Authors
The authors have improved the manuscript considerably. I would only suggest two small changes. The first relates to the frequency bands. Those included in this MS are the correct ones and it would be appropriate to cite Kuwahara et al (1996 Assessment of autonomic nervous function by power spectral analysis of heart rate variability in the horse. Journal of the autonomic nervous system, 60(1-2), 43-48). In terms of discussion, the authors should consider the possibility that the results obtained are also influenced by other variables, such as the difference in socialisation of the horses between the 3 stables. This strengthens the result by providing guidance to other researchers and shows that the authors have sufficient knowledge of other possible variables. A few lines in the discussion are sufficient.
Response to the reviewer
We’ve added the citation of Kuwahara et al (1996) in the 3rd paragraph of “2.4.3 Heart rate and heart rate variability” on page 7, lines 225-226 accordingly. Regarding the suggestion over the different socialisation issues, we agree with the reviewer that the differences in socialisation may impact distinctively on the HRV variables, and the result should be interpreted with caution. So, we decided to put it as the additional study’s limitation on page 18, line 566.

Reviewer 2 Report
Comments and Suggestions for Authors
The research is of interest to those housing horses in tropical, savannah climates and the authors measured multiple parameters to differentiate the local environment in 3 differently designed stables. I appreciate the additional clarity of the stable design that included the size of screen. Some concerns regarding the differences between the composition of groups, in number of horses and sex, particularly with the rationale of including 1 stallion, is a concern, as they tend to more alert and responsive to their environment than geldings. The uneven number of horses per treatment (housing) was different as well. From a biosecurity point of view, I understand the rationale of maintaining horses in their home stables, this does present a discussion item that has not been addressed. A cross-over design, or Latin Square would have been a better design scientifically. Additionally, the differences in floor slope and amount of bedding should be discussed in more detail as they may play a role in the authors’ findings, particularly with gas measurements. The authors should add discussion on the location of the boxes used in the study and how their locations relative to the ‘gates’ may influence some of the measured parameters. Boxes in stable C are located all in the interior of the barn, while boxes in stable A had 3 out of 8 located new the external gates. The authors should add information on the width of the aisles as this would affect airflow. The heart rate data is still presented in a cumbersome manner and some of the graphs are extremely crowded and difficult to interpret.
In some places the wording and sentence structure could be improved. Several previous suggestions to improve clarity and write more concisely were not incorporated in version 2. Overall, the manuscript is fairly well done and the evidence appears to support the author’s conclusions.

Comments on the Quality of English LanguageThere are places where the sentence structure could be improved. Authors should strive to provided details in the most concise manner.
Author Response
Reviewer 2
Title: Stress responses in horses housed in different stable designs during summer in a tropical savanna climate
Comments and Suggestions for Authors
The research is of interest to those housing horses in tropical, savannah climates and the authors measured multiple parameters to differentiate the local environment in 3 differently designed stables. I appreciate the additional clarity of the stable design that included the size of screen. Some concerns regarding the differences between the composition of groups, in number of horses and sex, particularly with the rationale of including 1 stallion, is a concern, as they tend to more alert and responsive to their environment than geldings. The uneven number of horses per treatment (housing) was different as well. From a biosecurity point of view, I understand the rationale of maintaining horses in their home stables, this does present a discussion item that has not been addressed. A cross-over design, or Latin Square would have been a better design scientifically. Additionally, the differences in floor slope and amount of bedding should be discussed in more detail as they may play a role in the authors’ findings, particularly with gas measurements. The authors should add discussion on the location of the boxes used in the study and how their locations relative to the ‘gates’ may influence some of the measured parameters. Boxes in stable C are located all in the interior of the barn, while boxes in stable A had 3 out of 8 located new the external gates. The authors should add information on the width of the aisles as this would affect airflow. The heart rate data is still presented in a cumbersome manner and some of the graphs are extremely crowded and difficult to interpret.
In some places the wording and sentence structure could be improved. Several previous suggestions to improve clarity and write more concisely were not incorporated in version 2. Overall, the manuscript is fairly well done and the evidence appears to support the author’s conclusions.
Response to the reviewer
We’d like to thank the reviewer for these valuable comments and provide more precise information regarding the experimental design in this study.
Age and sex are believed to cause differences in HRV modulation in horses. It’s clear that the age of the horse exerts an impact on HRV variables, as horses with advanced age show a reduced HRV compared to young horses (https://doi.org/10.1016/j.jveb.2020.05.005, https://doi.org/10.1294/jes.28.99). Although sex is thought to affect HRV modulation, Reitmann et al. (2004) could not determine gender-related HRV measures in horses (https://doi.org/10.1016/j.applanim.2004.02.016). However, homogenous or balanced groups were recommended. (https://doi.org/10.1016/j.jveb.2013.02.003, https://doi.org/10.1016/j.Physb
eh.2014.01.024, https://doi.org/10.1016/j.jevs.2024.105094). So, we focus on selecting horses with similar age and weight ranges in each stable design, resulting in no difference in age and weight among horses in the 3 stable designs (this has already been mentioned on page 11, lines 297-299).
As per the reviewer's concern over the inclusion of stallions in this study, there were no outlier data following the identified outlier analysis provided by the Graphad program, meaning that data from all 23 horses were included. Moreover, this one stallion is calm, behaves in no aggressive manner and is long-term housed in the box between 2 boxes of geldings. We believe the stallion condition exerts no effect on HRV variables but the housing condition itself. Despite the uneven number of horses in each housing, we produced a reliable result using the latest version of the GraphPad Prism, which provides plenty of appropriate statistical analysis methods, including a mixed-effects model (REML). This statistical method has the potential to estimate statistical significance in data analysis if some values are missing and there is an uneven number of samples in the comparison groups. (https://www.graphpad.com/guides
/prism/latest/statistics/stat_anova-approach-vs_-mixed-model.htm).
Regarding the study design, changing or swapping the housing of each horse with respect to cross-over designs may cause more trouble in producing HRV in each horse. As horses were measured for the RR interval data during housing in their stable, the variation in heart rates of each was supposed to reflect the real stress response from the normal housing. However, with a cross-over design study, the stress response may be overestimated following anxiety and fear when they were swapped to accommodate within an unfamiliar stable. That’s why we conducted the experiment by collecting data on horses housed in their stables.
The location of horses in stables A and B is supposed to have no effect on HRV modulation, as no outlier of HRV data was removed from horses in these two stables. Moreover, the 3-4 metres wide aisle and the opening gates throughout the day allow natural air to flow through the stable, supposing that all horses in stables A and B were exposed to air similar to their neighbouring boxes in each stable. Regarding the design of stable C in which no external solid results in free air flowing into the whole stable, assuming that all horses were equally exposed to the air regardless of the location of the boxes.
Concerning the differences in floor slope and amount of bedding, we’ve discussed the possible cause of early detection of ammonia levels in the stable in terms of floor slope (which is lower than the other 2 stables) and the bedding amount (which is less straw bedding in stable C than in other stables. This allowed such fluids to flow slowly and be absorbed into the concrete floor, facilitating ammonia emission within the boxes) on page 17, lines 521-527.

Reviewer 3 Report
Comments and Suggestions for Authors
I accept the corrections made.
Author Response
Reviewer 3
Title: Stress responses in horses housed in different stable designs during summer in a tropical savanna climate
Comments and Suggestions for Authors
I accept the corrections made.
Response to the reviewer.
We’d like to thank the reviewer for your dedication to reviewing and accepting the current form of our manuscript.
